# Novel Mg-0.5Ca-xMn Biodegradable Alloys Intended for Orthopedic Application: An In Vitro and In Vivo Study

**DOI:** 10.3390/ma14237262

**Published:** 2021-11-27

**Authors:** Corneliu Munteanu, Daniela Maria Vlad, Eusebiu-Viorel Sindilar, Bogdan Istrate, Maria Butnaru, Sorin Aurelian Pasca, Roxana Oana Nastasa, Iuliana Mihai, Stefan-Lucian Burlea

**Affiliations:** 1Mechanical Engineering Department, “Gheorghe Asachi” Technical University of Iasi, 700050 Iasi, Romania; corneliu.munteanu@academic.tuiasi.ro (C.M.); nastasa_ro@yahoo.com (R.O.N.); 2Technical Sciences Academy of Romania, 26 Dacia Blvd., 030167 Bucharest, Romania; 3Faculty of Medical Bioengineering, “Grigore T. Popa” University of Medicine and Pharmacy from Iasi, 9-13 Kogălniceanu Str, 700454 Iasi, Romania; mariabutnaru@yahoo.com; 4TRANSCEND Research Centre, Regional Institute of Oncology, Str. G-ral Henri Mathias Berthelot 2-4, 700483 Iasi, Romania; 5Faculty of Veterinary Medicine, “Ion Ionescu de la Brad” Iasi University of Life Sciences, 8, Mihail Sadoveanu Alley, 700490 Iasi, Romania; passorin@yahoo.com (S.A.P.); iuliabogdan2005@yahoo.com (I.M.); 6Faculty of Dentistry, “Grigore T. Popa” University of Medicine and Pharmacy from Iasi, 9-13 Kogălniceanu Str, 700454 Iasi, Romania; lucianburlea@yahoo.com

**Keywords:** Mg-Ca-Mn biodegradable alloys, in vitro analysis, in vivo analysis

## Abstract

Mg-based biodegradable materials, used for medical applications, have been extensively studied in the past decades. The in vitro cytocompatibility study showed that the proliferation and viability (as assessed by quantitative MTT-assay—3-(4,5-dimethyltiazol-2-yl)-2,5-diphenyl tetrazolium bromide) were not negatively affected with time by the addition of Mn as an alloying element. In this sense, it should be put forward that the studied alloys don’t have a cytotoxic effect according to the standard ISO 10993-5, i.e., the level of the cells’ viability (cultured with the studied experimental alloys) attained both after 1 day and 5 days was over 82% (i.e., 82, 43–89, 65%). Furthermore, the fibroblastic cells showed variable morphology (evidenced by fluorescence microscopy) related to the alloy sample’s proximity (i.e., related to the variation on the Ca, Mg, and Mn ionic concentration as a result of alloy degradation). It should be mentioned that the cells presented a polygonal morphology with large cytoplasmic processes in the vicinity of the alloy’s samples, and a bipolar morphology in the remote region of the wells. Moreover, the in vitro results seem to indicate that only 0.5% Mn is sufficient to improve the chemical stability, and thus the cytocompatibility; from this point of view, it could provide some flexibility in choosing the right alloy for a specific medical application, depending on the specific parameters of each alloy, such as its mechanical properties and corrosion resistance. In order to assess the in vivo compatibility of each concentration of alloy, the pieces were implanted in four rats, in two distinct body regions, i.e., the lumbar and thigh. The body’s reaction was followed over time, 60 days, both by general clinical examinations considering macroscopic changes, and by laboratory examinations, which revealed macroscopic and microscopic changes using X-rays, CT(Computed Tomography), histology exams and SEM (Scanning Electron Microscopy). In both anatomical regions, for each of the tested alloys, deformations were observed, i.e., a local reaction of different intensities, starting the day after surgery. The release of hydrogen gas that forms during Mg alloy degradation occurred immediately after implantation in all five of the groups examined, which did not affect the normal functionality of the tissues surrounding the implants. Imaging examinations (radiological and CT) revealed the presence of the alloy and the volume of hydrogen gas in the lumbar and femoral region in varying amounts. The biodegradable alloys in the Mg-Ca-Mn system have great potential to be used in orthopedic applications.

## 1. Introduction

Magnesium-based biodegradable alloys have the advantage of eliminating the second surgical intervention in the human body. The major disadvantage of these types of materials is that the biodegradation process is accelerated, which leads to decreased mechanical properties and inflammatory reactions under the skin with the release of hydrogen [1,2]. In the industrial domain, several discoveries have been made involving magnesium-based alloys, but the pure magnesium alloy presents high activity in an aqueous environment, low formability [3,4], low mechanical strength and the low precipitation of solid solutions, due to its hexagonal crystalline structure [4]. These disadvantages could lead to implant mass integrity loss or the excessive release of H_2_, which can decrease cellular development and can appear in local cysts [5]. Despite these aspects, alloying magnesium with biocompatible elements could replace permanent implants such as stainless steel, titanium-based alloys or Co-Cr-based alloys [6,7,8,9].

The first studies in this field were conducted in the early 20th century, where crystalline magnesium was used as a biomaterial, but were stopped due to the massive release of subcutaneous hydrogen [10]. In order to reduce the biodegradation rate, magnesium has been alloyed with Ca, Zn, Mn, Al or rare earth elements (RE), with calcium and zinc having the highest biocompatibility [11]. 

It is well known that calcium is a basic component in the bone structure, and it plays a very important role in its metabolism. In vitro analysis in cell culture media has shown that a high level of extracellular calcium stimulates the synthesis process in osteoblasts, developing their proliferative activity [12,13,14].

Rad et al. [15] presented, in their own research, the fact that the Ca concentration influences the formation of the lamellar eutectic compound Mg_2_Ca, which contributes to the decrease of corrosion resistance. On the other hand, studies have shown that manganese does not allow heavy metals’ incorporation, and contributes to their removal during the manufacturing process [16]. 

Manganese has no cytotoxic effects, and develops a major role in the activation of enzyme systems such as decarboxylases, kinases or hydrolases [17]. Mn contributes to the refinement of the microstructure and improves the yield strength of magnesium alloys. Studies by Khan et al. on the AZ31 and AZ10 alloys showed that the size of the Mg grains decreased with the increase of the Mn concentration up to 0.4 wt.% Mn [18,19].

Abdiyan et al. studied the degradation rate of GZ31 alloy alloyed with 1 wt.% Mn. It was observed that the grain size decreased from 3.7 µm to 2.3 µm, the hardness and mechanical strength were improved, and the degradation rate decreased from 2.1 mm/year (GZ31) to 0.2 mm/year (GZ31-1 wt.% Mn), due to the reduction of impurities [20].

Zhao et al. performed an in vivo analysis on three biodegradable alloy systems—AZ31 (an alloy containing 1 wt.% Mn), WKX41 and ZJ41—investigating the influence of alloy biodegradation in nine vital organs of rats. It was found that the level of chemical elements in these organs was not different from the control animal. Furthermore, the alloy with Mn (AZ31) had the lowest biodegradation rate among the three systems studied. One month after implantation, all three systems showed hydrogen releases [21].

Other in vitro and in vivo comparative studies were performed by Zhen et al. [22] on two Mg-3Sn-0.5Mn and WE43 alloys. The Mg-3Sn-0.5Mn alloy showed a higher corrosion potential (−1.563 V) than WE43 (−1.707 V), resulting in a lower degradation rate. In the case of cell viability using VSMC cell cultures (vascular smooth muscle cells), the alloys showed a similar behavior between 70% and 95% 72 h after co-incubation. In vivo analysis showed that the Mg-3Sn-0.5Mn has very good biocompatibility with the blood, having a hemolysis rate of only 0.128%, without identifying thrombocytes.

In this research paper, the authors studied a novel Mg–0.5 wt.% Ca experimental material which was alloyed with a controlled content of 0.5, 1.0, 1.5, 2.0 and 3.0 wt.% Mn in order to improve the biodegradability rate needed for a symmetrical balance between alloy degradation and bone healing, followed by in vitro 3-(4,5-dimethyltiazol-2-yl)-2,5-diphenyl tetrazolium bromide (MTT) tests to determine its biocompatibility. 

The tissular (soft tissue) reaction around the implanted material and biodegradation aspects were assessed using in vivo experiments and imaging diagnosis, i.e., RX (X-ray), CT, SEM, histopathology, and EDS (Energy Dispersive Spectroscopy) for evaluation at 1, 2, 4 and 8 weeks after implantation. The aim of the present work is to study the cellular viability and in vivo implantations, in order to highlight the alloy’s biocompatibility.

## 2. Materials and Methods

### 2.1. Elaboration of the Mg-Ca-Mn Alloys and Microstructural Analysis

The original experimental alloys were developed using master alloys of which the data are from Hunan China Co., Changsha, China, with percentages presented from other studies [23,24]. As in prior research [25], the Mg-based alloys were made by melting a square section of the original components in an induction furnace in an Ar-protected atmosphere, in a 675–700 °C temperature field for 30 min. The elaboration method offered cylindrical mini-ingots (with a diameter of 20 mm and a height of 50 mm), which were cut into smaller round plates with distinct chemical compositions depending on their Mn element concentration discrepancies, as indicated in Table 1. The circular samples have a diameter of 20 mm and a thickness of 2 mm. Furthermore, the material amount in grams was calculated in order to obtain ingots with a net mass of roughly 23 g per melt. The average chemical composition was measured in five different areas, and is presented in Table 2. After metallographic grinding with paper disks of 200–2500 MPi granulation, the specimens were mechanically polished with an Al_2_O_3_ suspension solution (2–5 µm) for the experimental tests. Chemical etching with Mg(CH_3_COO)2Mg_4_H_2_O acetate solution revealed the microstructure after 30 min of cleaning with ethyl alcohol.

Scanning electron microscopy (SEM) (FEI, Brno, Czech Republic) was used to examine the microstructural features (FEI Quanta 200-3D, FEI, Brno, Czech Republic). An energy-dispersive spectroscopy (EDS) detector (Ametek, Unterschleissheim, Germany) was used to determine the chemical makeup (Xflash, Bruker, Germany).

### 2.2. In Vitro Cytocompatibility Study

#### 2.2.1. Cell Culture

Albino rabbit dermal primary fibroblasts (passage No. 3) were used in this study as a largely studied cellular model, thanks to the fibroblast’s key contribution to post-injury new tissue formation (i.e., new collagen deposition, healing facilitation). The cells were cultured in a complete Dulbecco’s modified Eagle medium/Nutrient F-12 Ham (DMEM-F12 Ham) culture medium; i.e., cell culture media supplemented with 10% inactivated fetal bovine serum (FBS) and 1% antibiotic solution (5000 units penicillin/5 mg streptomycin/10 mg neomycin/mL), in a humidified atmosphere of 95% air and 5% carbon dioxide at 37 °C. The 90% confluent cells layer was rinsed with phosphate-buffered saline (PBS) solution, harvested by incubation with trypsin/ethylenediaminetetraacetic acid (Sigma Chemical Co., St. Louis, MA, USA), counted, and seeded in 24-well culture plates at a density of 1 × 10^4^ cells/well. 

After 24 h of incubation, the medium containing unattached or dead cells was removed by aspiration, the adhered cells layer was rinsed with PBS, and a fresh medium was added into each well. Subsequently, a well-permeable insert (with a membrane pore size of 0.4 μm) containing the studied Mg-alloy samples was inserted into the assigned wells of the abovementioned 24-well-plated cell cultures and co-incubated with the fibroblastic cells for 1, 3 and 5 days, in order to promote and sustain the Mg-alloy’s interaction with the cell culture media throughout the co-incubation periods for the cytocompatibility and cell morphology tests. In addition, negative controls were used for each time period, i.e., control-wells containing cell cultures only. The culture medium was refreshed every 2 days. 

#### 2.2.2. Cell Viability

The cytocompatible feature of the Mg100(n+x)-Ca(n)-Mn(x) alloys was assessed by the MTT (i.e., 3-(4,5-dimethyltiazol-2-yl)-2,5-diphenyl tetrazolium bromide) colorimetric assay [26,27,28], through which is quantified the metabolic activity of the live cells. Briefly, after specified periods (i.e., 1, 3 and 5 days) of the continuous exposition of the cells to 100% alloy extracts, the permeable inserts (containing the alloy samples) were taken out from the wells, and the cells were evaluated concerning their viability profile. 

For this purpose, the cells were washed with PBS (phosphate-buffered saline), treated with MTT dye solution diluted in fresh medium, and incubated at 37 °C for 3 h in order to assure the formazan crystal formation (as a dark-blue insoluble product) through the contribution of mitochondrial enzymes [26,27,28]. Afterward, formazan crystals were solubilized with isopropyl alcohol under continuous agitation (Environmental Shaker-Incubator ES-20, Biosan, Riga, Latvia) for 15 min, and the absorbance of the resulting liquid in each well was read by using a microplate reader (Tecan Sunrise, with Magelan V.7.1 soft for data acquisition, Tecan Group Ltd., Männedorf, Switzerland) at a wavelength of 570 nm. The cell viability (CV) results were expressed as a percentage of the optical density (Od) data of the control-wells according to the following formula: CV = 100 × (Od of alloy wells-Od of empty wells)/(Od of control wells-Od of empty wells). The statistical analysis of the cell viability data was performed using the one-way ANOVA test, and the obtained results were compared by applying Tukey’s method, with statistically significant differences being accepted at *p* < 0.05.

#### 2.2.3. Cell Morphology

The cells’ morphology was assessed through fluorescence microscopy performed after 1 and 5 days of the co-incubation of the fibroblastic cells with the alloy samples under the abovementioned conditions. For this purpose, the cells previously washed with Hanks’ Balanced Salt solution (HBSS; H8264, Sigma-Aldrich, Taufkirchen, Germany) without red phenol were treated with 200 µL/well of a 1:1000 Calcein AM solution (Calcein AM; C1359, Sigma-Aldrich, Taufkirchen, Germany) in HBSS, and were incubated for 30 min at 37 °C in a dark environment, in order to assure the intracellular conversion of the cell-permeant dye to a green-fluorescent calcein inside of the viable cells. Afterward, the cellular morphology and distribution were evaluated using an inverted microscope (Leica DMIL LED equipped with a Leica DFC450C camera and Leica Application Suite-Version 7.4.1 image acquisition software, Leica, Wetzlar, Germany).

### 2.3. In Vivo Animal Study

The implants used in this study were alloys with a majority content of Mg, 0.5% calcium (Ca), and manganese (Mn) in varying amounts. The codification of the alloys is the following: M1 for Mg0.5Ca0.5%Mn, M2 for Mg0.5Ca1.0%Mn, M3 for Mg0.5Ca1.5%Mn, M4 for Mg0.5Ca2.0%Mn, and M5 for Mg0.5Ca3.0%Mn. The implants used had a parallelepiped shape, with rounded corners, 10–13 mm long and 1.5–2 mm high, similar to other studies [29]. We calculated the dimensions of the implants using a specific orthopaedic algorithm. The weight of the animal is directly proportional to the size of the long bones. The length of the implant was set at one third of the length of the femur, and the thickness of the implant was calculated taking into account the average diameter in the middle third and the fact that it must not exceed 60% of the bone diameter. The animal experiments were performed in compliance with the Guide for Care and Use of Laboratory Animals and the European legislation on animal use. Male Sprague Dawley rats weighing 250–350 g and 8 months of age were randomized into five groups of 4 animals each. A piece of alloy was implanted in the femoral region (Figure 1a) and then the lumbar region (Figure 1b) of each rat (Figure 1c).

Preoperatively, the rats were given an injection i.m. with xylazine, for the installation of general anesthesia and its maintenance throughout the operation by the masked inhalation of a mixture of 1–2% isoflurane and oxygen. The surgical area was shaved and disinfected with betadine solution. The operative access was ensured by positioning the rats on the operating table in a sterno-abdominal lying position. The alloy pieces were implanted using a sharp 3–4 cm incision of the skin, followed by blunt dissection to tear the muscles on the bone surfaces. The muscular layer was sutured in a continuous fashion, and the skin was sutured in separate points. Postoperatively, the rats were housed in cages and allowed to move freely, being monitored daily for the visual assessment of their mobility. 

Computed radiographs were performed with sedation at 7, 14, 30, and 60 days after surgery in order to obtain digital diagnostic images interpreted directly on a computer monitor using the Examion XCR Smart (Agfa Healthcare, N.V. Belgium). The tissue reaction was monitored by histological examination using trichrome Masson staining, and was examined under a photonic microscope using an Olympus imaging and processing system, namely an Olympus^®^ BX51 microscope (Olympus Life Science GMBH, Hamburg, Germany) and the Olympus Cell B imaging program. 

All of the animals in the groups subjected to surgical experimentation were euthanized according to a well-established protocol, such that, from beginning to end, we can assess the degree of biocompatibility, and also the resorption of the implanted material. In order to obtain surface and chemical information, the samples were fixed in 2% glutaraldehyde solution in phosphate buffer, dehydrated in successive baths of alcohol (alcohol 30%, 50%, 70%, 90%, absolute alcohol and acetone), dried freely in the air, and finally covered with gold. The microstructural, morphology and EDS analysis were carried out by scanning electron microscopy (SEM FEI Quanta 200 3D, dual beam, FEI, Brno, Czech Republic, equipped with energy dispersive X-ray spectroscopy analysis unit—Xflash Bruker, Harvard, MA, USA).

## 3. Results and Discussion

### 3.1. Microstructural Characterization

Figure 2 highlights the SEM microstructures of the tested alloys. The structure of Mg-0.5Ca-xMn alloys consists of polyhedral grains of the alpha-Mg type, a Mg_2_Ca lamellar compound identified at the grain boundary with a higher concentration of Ca. The microstructure images show the presence of uniformly distributed white globular particles, enriched in Mn, as have been identified in other studies [30]. 

The XRD analysis shown in Figure 3 and the light microscopy performed by Oprisan, Munteanu, Istrate et al. [31] on the same set of samples presented three major phases: α-Mg, Mg_2_Ca and Mg_0.975_Mn_0.025_. It was shown that Mg_0.975_Mn_0.025_ has the same hexagonal crystallographic structure with pure Mg, and that Mg_2_Ca has a monoclinic structure.

The calculation of the average grain sizes of the Mg-0.5Ca-xMn alloys was performed for a minimum of seven areas. It was observed that a substantial decrease in grain size appeared due the increase in the concentration of Mn in the Mg-0.5Ca-xMn alloys. The values are highlighted in Table 3.

The uniform distribution of Mg grains and Mn-rich particles is highlighted in Figure 4. In addition to the enriched globular particles, Mn is found in the metal mass dissolved in a solid solution, which influences the average size of the grains.

### 3.2. Cell Viability

Figure 5 shows the results of the MTT test applied for the assessment of the cytocompatibility of the Mg100(n+x)-Ca(n)-Mn(x) alloys, expressed as percentages from the viability data obtained for the control wells (i.e., the negative control). It can be observed that after 24 h of the co-incubation of the cells with the studied alloy samples, the level of cell viability was maintained at similar values for all of the samples (*p* > 0.05), except for the MgCa alloy, in which case the level of cell viability was significantly lower than in the case of the other alloys (*p* < 0.05). 

The test performed after 3 days of co-incubation showed similar levels of viability in the case of alloys with a quantity of 0.5% Mn and 2% Mn (*p* > 0.05), but significantly higher (*p* < 0.05) compared to the other alloys. The cell viability recorded after 5 days reached a similar level (i.e., there is no statistically significant difference between the Mg100(n+x)-Ca(n)-Mn(x) alloys studied, *p* > 0.05), except for the MgCa alloy, in which case the level of cell viability was significantly higher than in the case of all the other alloys (*p* < 0.05).

It should be pointed out that for the cytocompatibility study among the studied alloys, the native MgCa alloy was also included, in order to obtain comparative results on the effect of the alloying element on the cytocompatibility of the Mg100(n+x)Ca(n)Mn(x), as the well-known low chemical stability of the MgCa alloys in complex humid environments can be improved by alloying (i.e., culture medium or biological fluids [32,33], thereby improving the cytocompatibility properties. In addition, these facts are supported by our previously published data [31] concerning the electro-corrosion resistance of the same studied alloys, showing that the corrosion resistance increased with the amount of Mn as an alloying element. Briefly, the electrochemical studies performed on the same set of samples [31] evidenced that the highest corrosion rate resulted for the alloy with 0.5% Mn for 0.85 mm/year, and the smallest corrosion rate was found for 3% Mn with 0.55 mm/year, leading to a decrease in the biodegradation rate.

In this sense, the results of the cell viability test (Figure 5) could be explained by the fact that alloying with Mn increased the chemical stability of the Mg100(n+x)Ca(n)Mn(x) alloys (compared with the MgCa alloy), and therefore the degradation process was limited, thus both the massive release of hydrogen (which can change the pH and subsequent alkalinization with the impaired cells’ metabolic activity) and the release of calcium ions and magnesium—which over certain concentrations may have a cytotoxic character (by increasing osmolarity, which can lead to hyperosmotic shock)—was avoided [34]. Moreover, the level of cell viability (Figure 5) registered for the MgCa alloy after 1 day was very low (i.e., 39.48%), but after 3 days it was 72.61%, and after 5 days it reached 101.75%, a fact that leads us to appreciate that the disintegration/degradation of the MgCa alloy occurs mainly in the first stage of contact with the culture medium. In addition, the cell viability recorded both after 1 day (84.82–89.65%) and after 5 days (82.43–86.20%) of co-incubation reached a similar level for all of the studied Mg100(n+x)Ca(n)Mn(x) alloys (*p* > 0.05), which could indicate that only 0.5% Mn is enough to improve the chemical stability and thus the alloy’s cytocompatibility. These results allow us to conclude (according to the ISO 10993-5 standard [35]) that the studied Mg100(n+x)Ca(n)Mn(x) alloys do not have long-term cytotoxic features, due to the fact that the viability tests were carried out by exposing the cells to the entire set of events that occurred during the Mg alloys’ degradation (i.e., 100% extracts, as the metal samples were suspended in the wells by means of inserts, for up to 5 days), which involves the release of hydrogen by the reduction of H_2_O to H_2_ gas, OH^−^ ions and Mg(OH)_2_ as a corrosion product [31].

### 3.3. Cell Morphology

Figure 6 shows the density and morphology of the cells, co-incubated with the studied Mg-based alloys for 1 and 5 days, as assessed by fluorescence microscopy. The different morphology of the cells was observed depending on their location in relation to the insert’s membrane. Thus, cells with an elongated appearance (bipolar morphology) are observed in the areas with high cell density; however, towards the center of the well and near to the insert membrane (where the cell density is lower) the cells have a polygonal morphology with extensive lamellipodia-like cytoplasmic processes. In addition, after 1 day, the cell density recorded for the Mg100(n+x)Ca(n)Mn(x) alloys was lower than in the case of the control wells (containing only cells without a metal sample) but higher than in the case of the MgCa alloy. These observations were found, to some extent, after 5 days of co-incubation, but this time the cell density was higher in the case of the MgCa alloy compared to the Mg100(n+x)Ca(n)Mn(x) alloys.

Furthermore, the results of the cell viability tests are well correlated with the results on the density and morphology of the fibroblastic cells (Figure 6). In this sense, these variations in cell viability, density and morphology could be attributed to the variations in the Ca, Mg and Mn concentrations in the microenvironment of the well, it being well known that the structural role of calcium is related to the reorganization of the cytoskeleton elements [36], along with the inhibitory effect of increased concentrations of Mg ions [37].

### 3.4. In Vivo Clinical Results

The Mn alloy implants’ tissue reactions were observed using general semiology clinical examination methods. All of the animals that underwent surgery were assessed for signs of local or systemic reactions every day. Hydrogen gas release occurred immediately after implantation in all of the studied alloys in both anatomical regions (Table 4). During the 60 days, large (L), medium (M) and small (S) tissue reactions, as well as their absence (A), were noticed.

The day after the operation, a local peri-implant reaction of variable intensity was observed in all five groups, in both body regions, due to the release of hydrogen gas formed during the degradation of the Mn alloy. The formed gas bags visibly changed the anatomical shape. Large accumulations of gas with deformations of the region were observed in groups M3, M4 and M5, but this did not influence the normal functionality of the tissues surrounding the implant.

### 3.5. Imagistic Interpertation

Computed radiographs were performed to show the morphological changes associated with the clinical symptoms. The X-rays (Figure 7) showed an increase in the volume of the regions, an increase in radiopacity due to the accumulation of gas in the soft tissues adjacent to the implant, hydrogen bags of varying sizes, and the presence of the implanted alloys in both dorsoventral presentation (7a, c, e and g) and lateral presentation (6b, d, f and h). Large bags of hydrogen gas were observed in the first 7 days after surgery in all five groups, both around the alloy and in the adjacent tissues.

The CT imaging investigations (Figure 8) performed at 7 (8 a, b), 14 (8 c, d), 30 (8 e, f) and 60 (8 g, h) days showed the presence of the alloys and hydrogen gas bags in the soft tissues adjacent to all of the groups of rats, in both regions. Compared to the first day, the gas bags decreased in volume such that, on the 60th day, they were no longer visibly deformed, with no visible changes in the volume of the region; small gas deposits were observed.

At 7 days, medium accumulations of gas with deformations of the region were observed in groups M2, M3, M4 and M5 (Figure 7a,b and Figure 8a,b), and small accumulations of gas were observed in group M1. At 14 days, medium accumulations of gas in the femoral region were observed in groups M4 and M5 (Figure 7c,d and Figure 8c,d), and small accumulations of gas were observed in groups M1, M2 and M3. At 30 days, small accumulations of gas with deformations of the region in groups M2, M4 and M5, and small accumulations of gas without deformation in groups M1 and M3 were observed (Figure 7e,f and Figure 8e,f). At 60 days, small accumulations of gas without deformations of the region were observed in all of the groups (Figure 7g,h and Figure 8g,h). Imagistic observations revealed the lowest amount of H2 for groups M1 and M3. 

### 3.6. Histological Analysis

The histological analysis (Figure 9) of the implanted areas at 7 (9a, b, c), 14 (9d, e, f), 30 (9g, h, i) and 60 (9j, k, l) days after the surgical intervention shows the very good resorption/biodegradation of the implanted material and a moderate inflammatory reaction.

The unformed connective tissue that gradually replaces the implanted material is well organized and well vascularized. The tissue biocompatibility of the material expressed good resorption and moderate local inflammation.

### 3.7. Scanning Electron Microscopy for the In Vivo Analysis

The SEM images for the Mg-0.5Ca-xMn alloys from the tissue samples at 7 (Figure 10), 14 (Figure 11), 30 (Figure 12) and 60 (Figure 13) days after surgery identified a cell field specific to the tissue reaction which shows a very good resorption/biodegradation of the implanted material and a moderate inflammatory reaction.

At 7 days after implantation, the SEM images showed cell adhesions to the material used. Macrophages, fibroblasts, and giant cells are visible in the fields. An intense neo-synthesis of collagen around the implantation region was also observed (Figure 10). At 14 days postoperatively, we further identified cell adhesions and the neoformation of collagen fibers. At the same time, we noticed the presence of inflammatory reactions, lymphocytes, and macrophages on the surface of the material used. The adhesion to the material of giant cells and macrophages involved in the process of resorption and proliferation of fibroblasts and the formation of collagen (Figure 11) was also observed. After a month (30 days), we also observed adhesions with giant cells and macrophages involved in the resorption of the material, proliferation with fibroblasts, and the synthesis of collagen fibers. From the images, we can conclude that connective tissue of neogenesis replaced most of the resorbed material (Figure 12). At 60 days, the SEM images show a new area of connective tissue that replaced the material, and thus the total resorption (disappearance) of the alloy (implant). (Figure 13).

The results of the histological and electro-microscopic examinations are identical, confirming the efficient bio-integration of the studied material. We can observe, upon examination after 60 days, that the number of collagen fibers is much higher compared to the period of 7, 14, and 30 days, indicating good biocompatibility, which ensures the recommendation of use in osteosynthesis/orthopedics. The large number of collagen fibers around the implant and the high cellularity (fibroblasts, giant Muller cells, macrophages, histiocytes, lymphocytes) indicate an intense resorption process and good tissue compatibility. No areas of peri-implant necrosis were noticed, which, in our opinion, indicates an excellent tolerance of the material, making it ideal for use as an osteosynthesis material.

## 4. Conclusions

The biodegradable Mg-0.5Ca-xMn alloy system was developed for validation in terms of biocompatibility and potential use in future orthopedic applications. The biodegradable Mg-0.5Ca-xMn alloy system was developed for the investigation of its biocompatibility and potential use in future orthopedic applications. From a microstructural point of view, the formation of polyhedral alpha-Mg grains was observed, and Mn was found to be dissolved in the solid solution, and in the form of enriched white globular particles.

The fact that the level of viability recorded both after 1 day (84.82–89.65%) and after 5 days (82.43–86.20%) of co-incubation reached a similar level for all of the studied Mg100(n+x)Ca(n)Mn(x) alloys, allows us to conclude that the studied Mg100(n+x)Ca(n)Mn(x) alloys do not have a long-term cytotoxic character. Moreover, the results seem to indicate that only 0.5% Mn is enough to improve the chemical stability, and thus the alloys’ cytocompatibility, and in this way could provide some flexibility in choosing the right alloy for a medical application depending on the specific mechanical and corrosion resistance parameters of each alloy. 

The clinical examinations found the presence of a local reaction of variable intensity for all of the implants, both in the lumbar region and in the femoral region. The degradation of Mg alloy implants in all five groups of rats caused an immediate release of hydrogen gas that did not affect the normal functionality of the tissues surrounding the implants. X-ray and CT imaging performed at 7, 14, 30 and 60 days after the surgery showed the presence of the alloys surrounded by gas deposits, as well as in the adjacent soft tissues for both regions. The gas volumes decreased, and 60 days after surgery small gas deposits were observed without visible changes in the body’s surface. The histological analysis of the implanted areas at 7, 14, 30 and 60 days after surgery showed a very good resorption/biodegradation of the implanted material and a moderate inflammatory reaction. Small gaps were observed around the implanted material due to gas accumulations caused by the resorption of the material.

The SEM images at 7, 14, 30 and 60 days after surgery identified cell adhesion and collagen fibers on the surface of the implanted material, demonstrating an efficient alloy integration in the tissue, the resorption of the implanted material, the neogenesis of connective tissue, the total resorption of the implanted material and its replacement by connective tissue.

It was noted, upon examination after 60 days, that the number of collagen fibers was much higher compared to the period of 7, 14 and 30 days, which indicates good biocompatibility and encourages the recommendation of use in osteosynthesis/orthopedics. The large number of collagen fibers around the implant and the high cellularity (fibroblasts, giant Muller cells, macrophages, histiocytes, and lymphocytes) indicate an intense resorption process and good tissue compatibility. No areas of peri-implant necrosis were noticed, which, in our opinion, indicates an excellent tolerance of the material, making it ideal for use as an osteosynthesis material

The current results confirmed that all of the magnesium alloys are biocompatible/biodegradable, and these properties recommend their use as possible materials for new devices.

## Figures and Tables

**Figure 1 materials-14-07262-f001:**
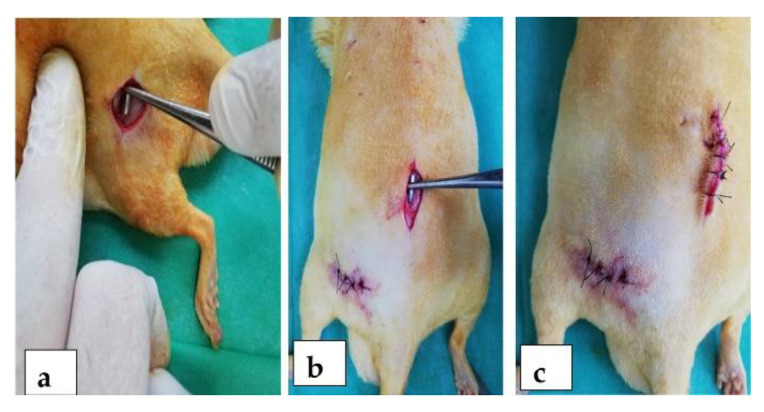
Rat with the alloy Mg-0.5Ca-xMn implanted in the femoral region (**a**) and then the lumbar region (**b**,**c**).

**Figure 2 materials-14-07262-f002:**
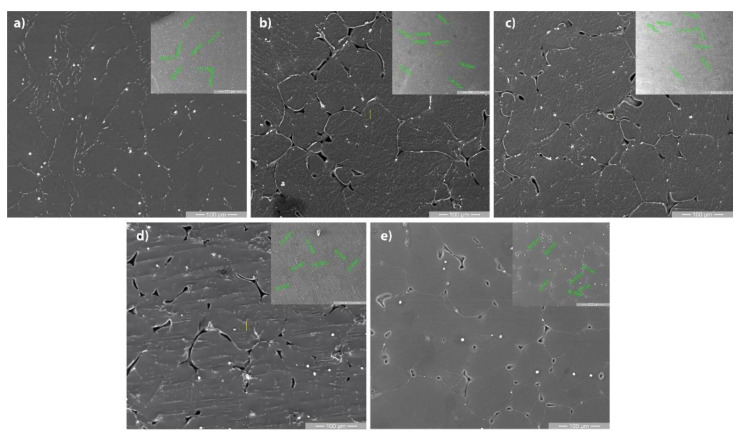
Scanning electron images of the Mg-0.5Ca-xMn alloys: (**a**) 0.5Mn; (**b**) 1Mn; (**c**) 1.5Mn; (**d**) 2Mn; (**e**) 3Mn.

**Figure 3 materials-14-07262-f003:**
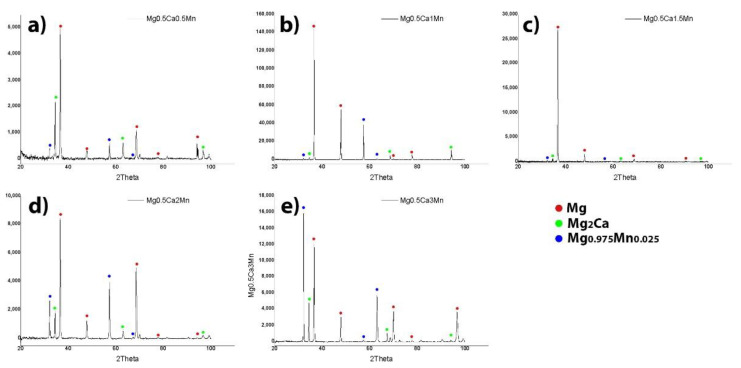
XRD diffraction patterns of the Mg-0.5Ca-xMn alloys: (**a**) 0.5Mn; (**b**) 1Mn; (**c**) 1.5Mn; (**d**) 2Mn; (**e**) 3Mn [31].

**Figure 4 materials-14-07262-f004:**
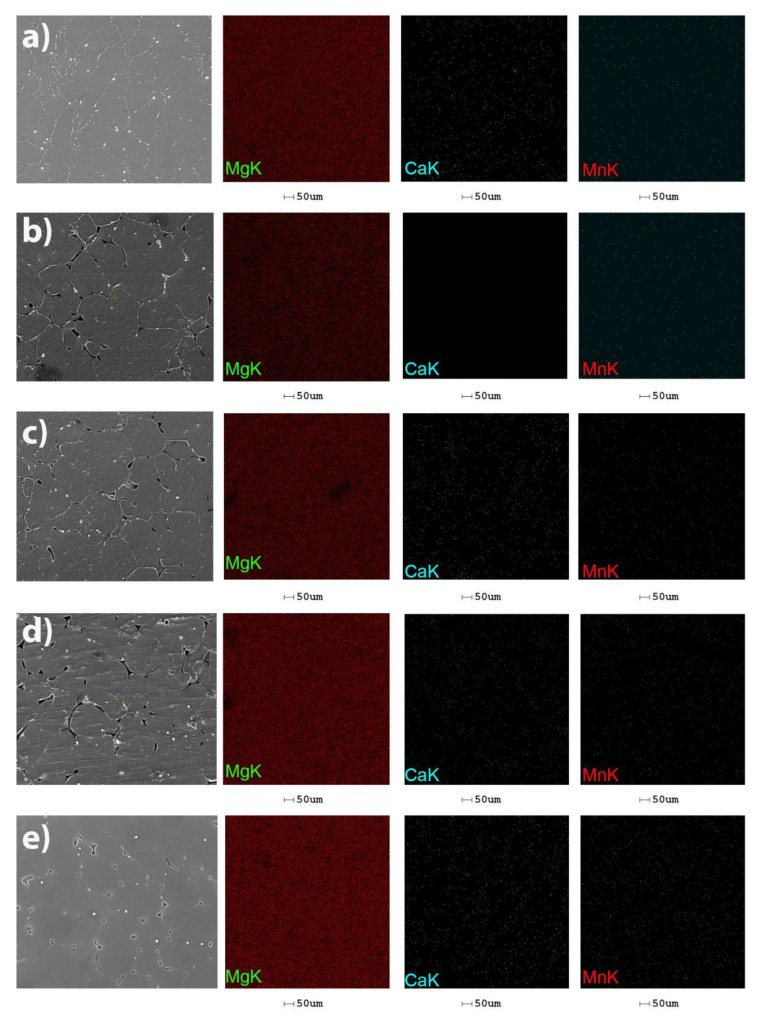
Distribution maps of the chemical elements in the Mg-0.5Ca-xMn alloys. (a) Mg-0.5Ca-0.5Mn; (**b**) Mg-0.5Ca-1Mn; (**c**) Mg-0.5Ca-1.5Mn; (**d**) Mg-0.5Ca-2Mn; (**e**) Mg-0.5Ca-3Mn.

**Figure 5 materials-14-07262-f005:**
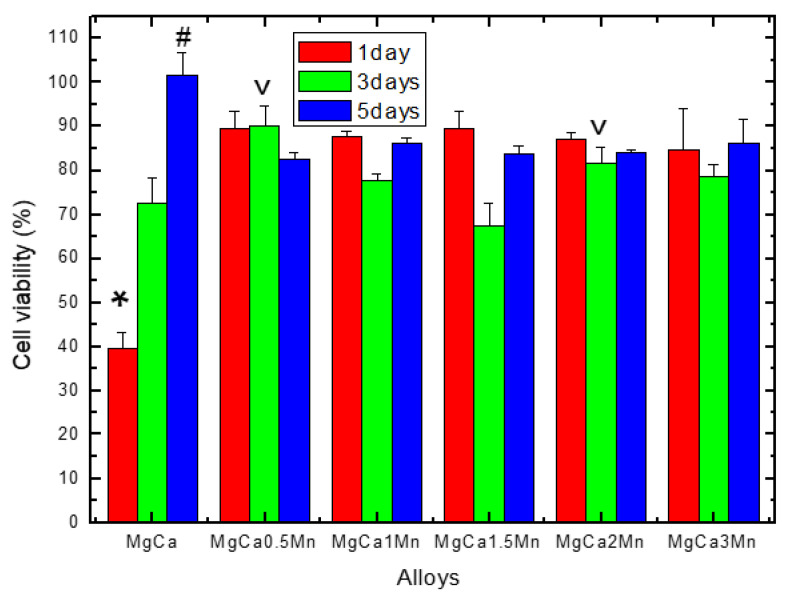
Cellular viability evaluated by an MTT assay; the effect of Mg100(n+x)-Ca(n)-Mn(x) experimental alloys on the fibroblastic cells’ viability after 1, 3, and 5 days of co-incubation. *, #, ˅ Statistically significant differences (*p* < 0.05) compared to the other alloys (see the text for details).

**Figure 6 materials-14-07262-f006:**
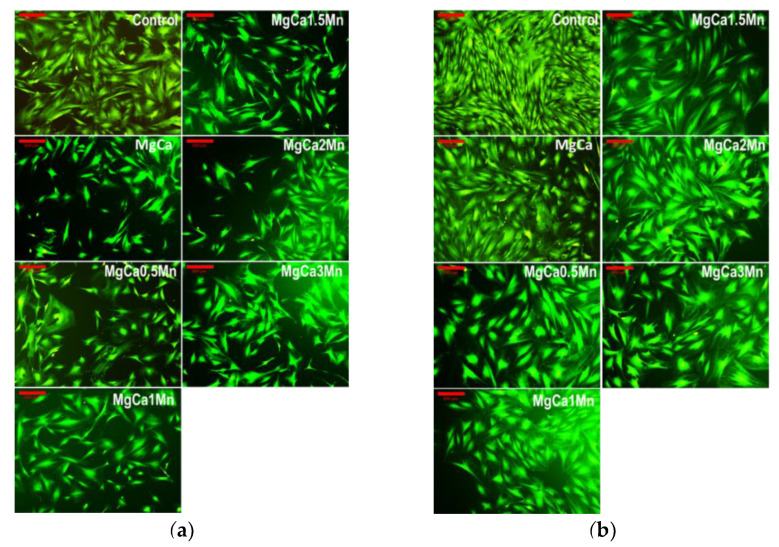
Fluorescence microscopy images showing the morphology of the viable cells (stained in green) after 1 (**a**) and 5 (**b**) days of co-incubation with the studied alloys (see the text for details). Bar: 200 μm.

**Figure 7 materials-14-07262-f007:**
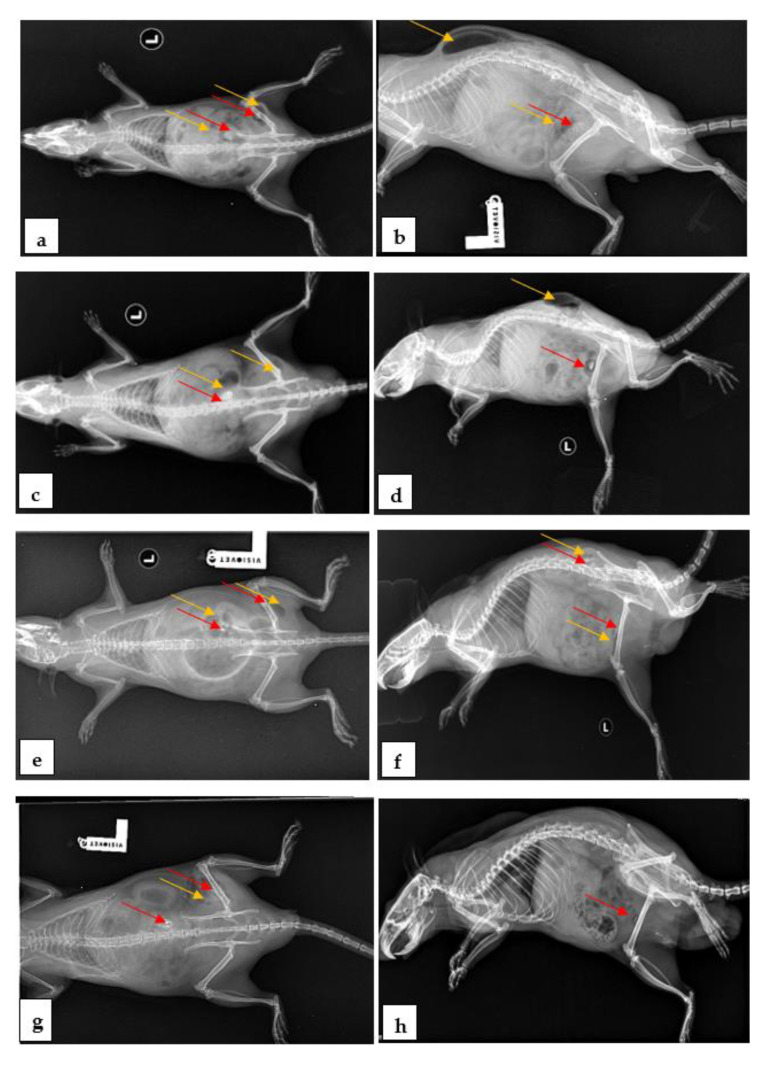
Rats: a representative X-ray image showing the gas deposit (yellow arrow) and alloy (red arrow). After 7 days postoperatively: (**a**) dorsoventral recumbency, (**b**) laterolateral recumbency. After 14 postoperative days: (**c**) dorsoventral recumbency, (**d**) laterolateral recumbency. After 30 postoperative days: (**e**) dorsoventral recumbency, (**f**) laterolateral recumbency. After 60 days postoperatively: (**g**) dorsoventral recumbency, (**h**) laterolateral recumbency.

**Figure 8 materials-14-07262-f008:**
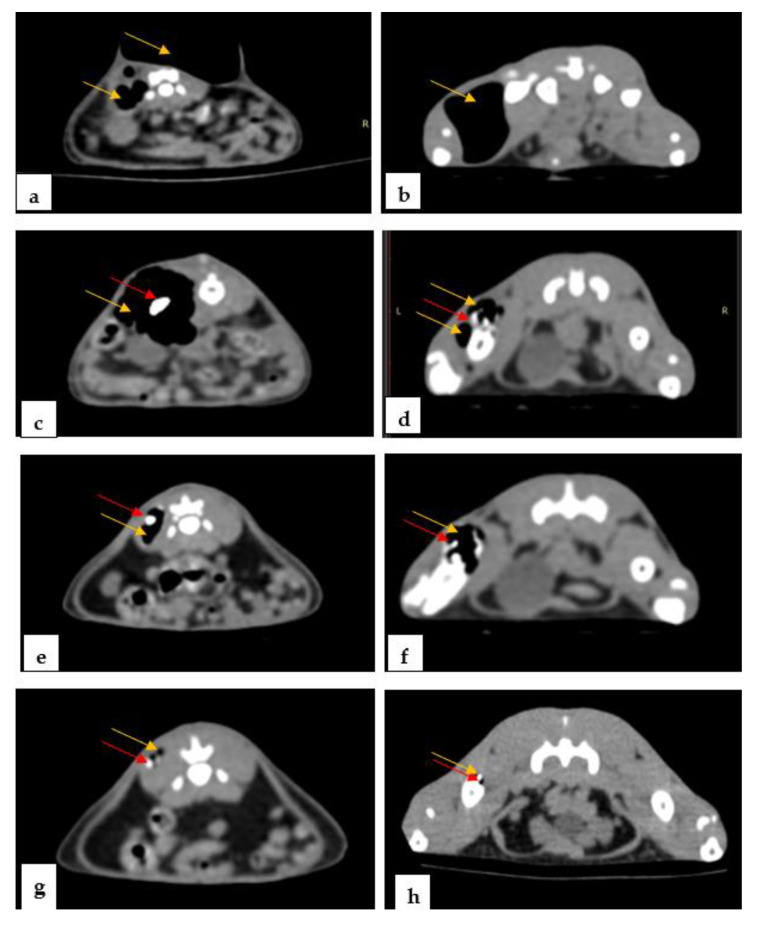
Rats: representative CT imaging appearance showing the gas deposits (yellow arrow) and alloy (red arrow). After 7 days postoperatively: (**a**) femoral and lumbar region, (**b**) femoral region. After 14 postoperative days: (**c**) lumbar region, (**d**) femoral region. After 30 postoperative days: (**e**) lumbar region, (**f**) femoral region. After 60 days postoperatively: (**g**) lumbar region, (**h**) femoral region.

**Figure 9 materials-14-07262-f009:**
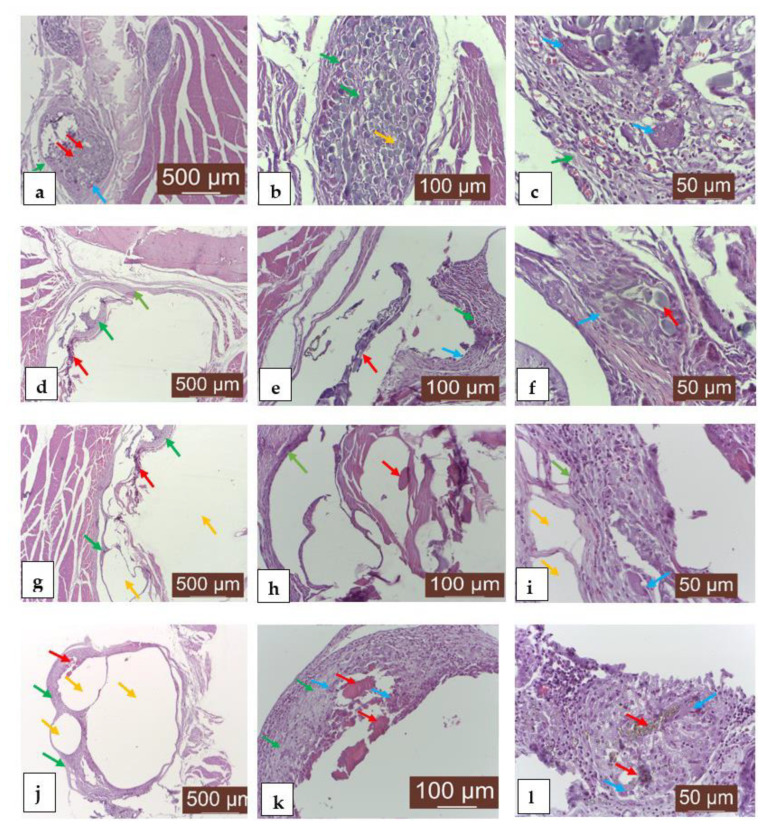
Representative results of the histological exams of MTC-stained sections of rat tissues after Mg-0.5Ca-xMn alloy implantation. Aspect after 7 days: (**a**) material implanted in large quantities (red arrow), under resorption, delimited by an area of new formed connective tissue-collagen fibers and fibroblasts (green arrow) and a reduced inflammatory reaction, represented by the presence of giant foreign-body cells (Muller cells), macrophages, rare lymphocytes and histiocytes (blue arrow); (**b**) connective tissue in the form of fine beams (green arrow) and a discrete inflammatory reaction, as noted by the presence of giant foreign-body cells (blue arrow); (**c**) post-implantation scarring area rich in giant foreign-body cells (blue arrow), scar connective tissue (green arrow), neoformation blood vessels, lymphocytes and histiocytes. Aspect after 14 days: (**d**) connective tissue neogenesis (green arrow) and the presence of the implanted material (red arrow); (**e**) small amounts of implanted material (red arrow) embedded in an new formed connective tissue (green arrow), giant cell inflammatory reaction (blue arrow); (**f**) giant cell (blue arrow) and fibrous inflammatory reaction (green arrow) around the implanted material (red arrow) for the purpose of biodegradation/resorption. Aspect after 30 days: (**g**) area of the resorption of the implanted material (red arrow) represented by connective tissue (green arrow) and gas bubbles (yellow arrow); (**h**) remains of the implanted material (red arrow), new-formed connective tissue (green arrow) surrounding the material, fibrous inflammation; (**i**) area of connective tissue (green arrow) and blood capillaries of neoformation, infiltrated with giant foreign-body cells, macrophages, lymphocytes, histiocytes (blue arrow) and gases (yellow arrow) in relatively small quantities, which tears the connective fibers of the connective tissue. Aspect after 60 days: (**j**) area of new-formed connective tissue (green arrow) enclosing a small amount of debris from the implanted material (red arrow) and gas bubbles (yellow arrow) resulting from the resorption of the material; (**k**) implanted material (red arrow) delimited by a discrete local inflammation with chronic evolution, consisting of a small population of cells, represented by macrophages, giant foreign-body cells, rare lymphocytes and histiocytes (blue arrow), and fibroblasts (green arrow); (**l**) reduced amounts of the implanted material (red arrow) with resilient crystalloid appearance, in contact with giant foreign-body cells and macrophages (blue arrow). Rare lymphocytes and histiocytes are noted.

**Figure 10 materials-14-07262-f010:**
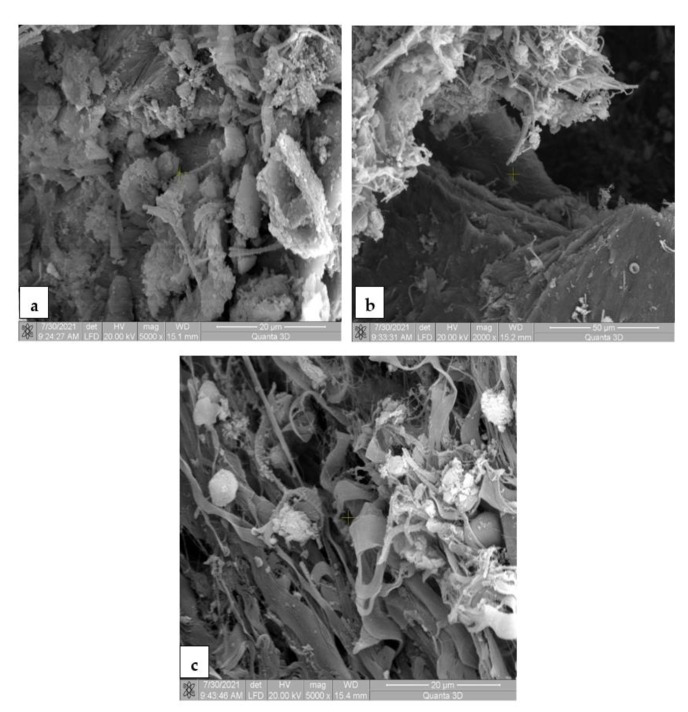
SEM images for biodegradable Mg-0.5Ca-xMn alloys in the tissue 7 days postoperatively, at a magnification of 5000× (**a**,**c**) and 2000× (**c**). (**a**) Cellular and fibrillar adhesion to the implanted material, new-formed collagen fibers, macrophages, fibroblasts, giant foreign-body cells; (**b**) collagenization area around the implanted material, collagen fibers and fibroblasts; (**c**) intense neosynthesis of the collagen fibers at the implantation site.

**Figure 11 materials-14-07262-f011:**
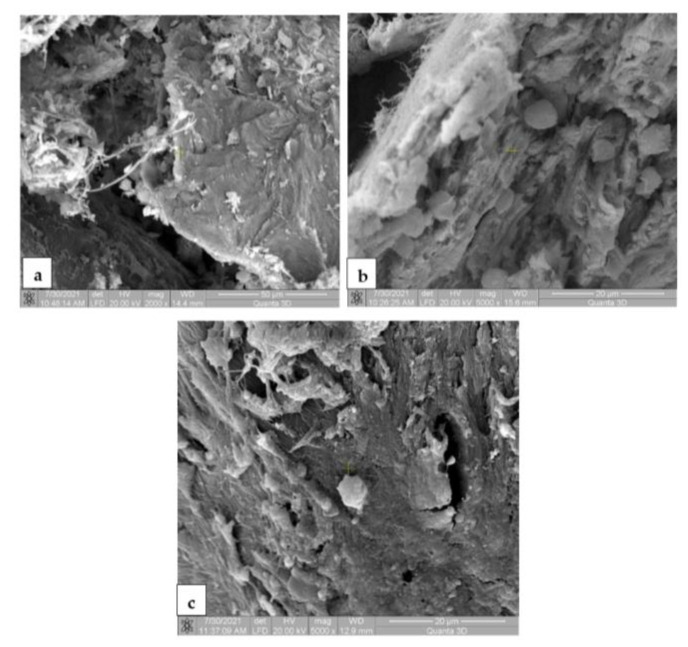
SEM images for biodegradable Mg-0.5Ca-xMn alloys in tissue 14 days postoperatively, at a magnification of 2000× (**a**) and 5000× (**b**,**c**). (**a**) Cell adhesion and collagen fibers on the surface of the implanted material, demonstrating efficient tissue integration, the resorption of the implanted material and the neogenesis of connective tissue; (**b**) inflammatory connective tissue and infiltrates (lymphocytes, macrophages) on the surface of the implanted material; (**c**) adhesion on the implant surface of giant cells and macrophages involved in the resorption and proliferation of fibroblasts and the synthesis of collagen fibers.

**Figure 12 materials-14-07262-f012:**
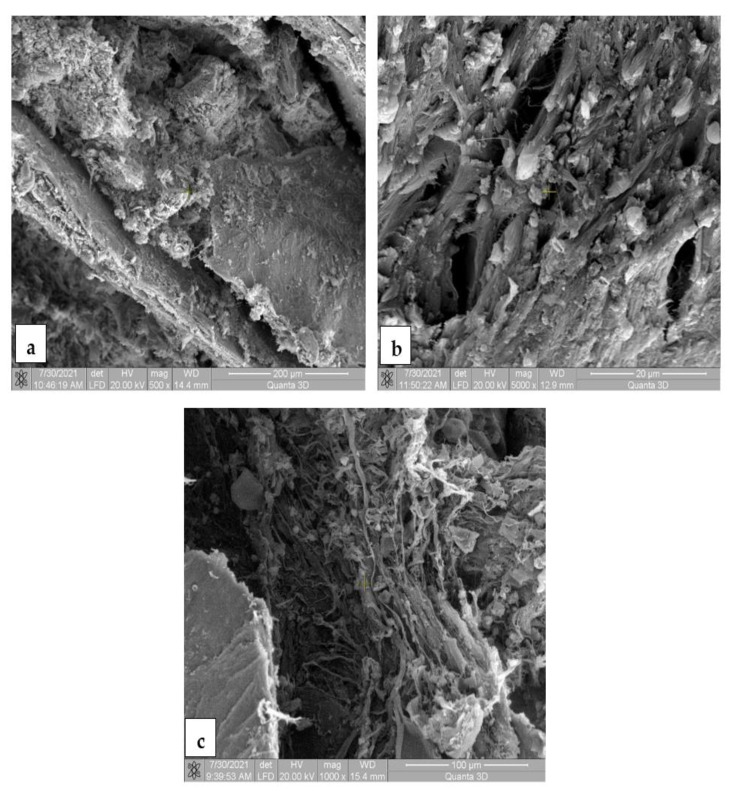
SEM images for biodegradable Mg-0.5Ca-xMn alloys in tissue 30 days postoperatively, at a magnification of 500× (**a**), 5000× (**b**) and 1000× (**c**). (**a**) Cell adhesion and collagen fibers on the surface of the implanted material, demonstrating efficient tissue integration, implanted material resorption and connective tissue neogenesis; (**b**) adhesion on the implant surface of giant cells and macrophages involved in the resorption and proliferation of fibroblasts and the synthesis of collagen fibers; (**c**) connective tissue (fibroblasts and collagen fibers) that replaces the resorbed material.

**Figure 13 materials-14-07262-f013:**
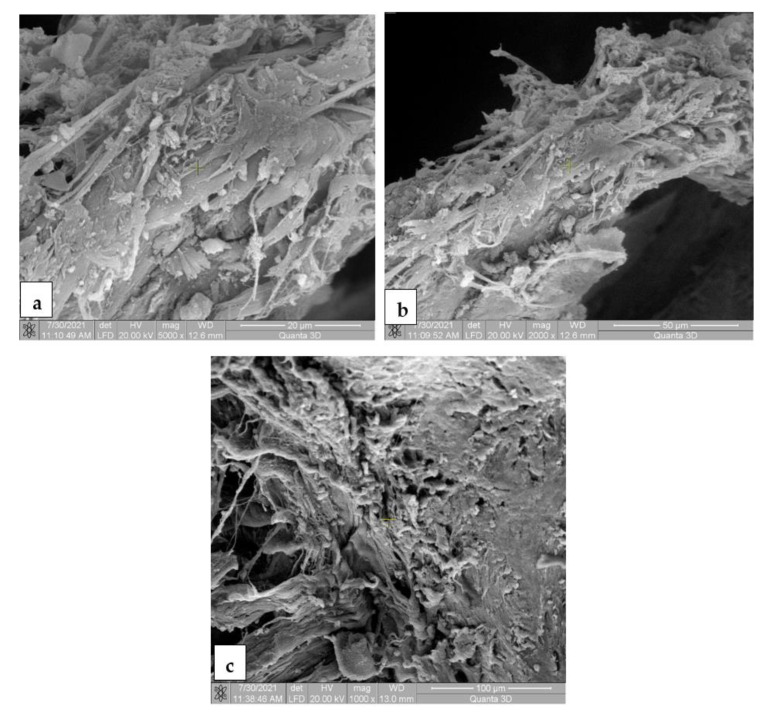
SEM images for biodegradable Mg-0.5Ca-xMn alloys in tissue 60 days postoperatively, at a magnification of 5000× (**a**), 2000× (**b**) and 1000× (**c**). (**a**) Resorption and integration of the implant in unformed connective tissue; (**b**) the area of semi-oriented connective tissue that replaces the resorbed material; (**c**) total resorption of the implanted material and its replacement by connective tissue.

**Table 1 materials-14-07262-t001:** Initial material masses used for the metal charges in order to obtain the experimental samples [23,24].

Specimens	Chemical Composition	Mg [g]	Mg–15Ca [g]	Mg–3Mn [g]
MgCa0.5Mn/M1	Mg0.5%Ca0.5%Mn	18.40	0.77	3.83
MgCa1Mn/M2	Mg0.5%Ca1%Mn	14.56	0.77	7.67
MgCa1.5Mn/M3	Mg0.5%Ca1.5%Mn	10.73	0.77	11.50
MgCa2Mn/M4	Mg0.5%Ca2%Mn	6.9	0.77	15.33
MgCa3Mn/M5	Mg0.5%Ca3%Mn	-	0.77	22.23

**Table 2 materials-14-07262-t002:** The average experimental chemical composition of the Mg-0.5Ca-xMn alloys, measured in five different areas.

Chemical Composition	wt.% Mg	wt.% Ca	wt.% Mn
Mg0.5%Ca0.5%Mn	98.89	0.56	0.55
Mg0.5%Ca1%Mn	98.18	0.83	0.98
Mg0.5%Ca1.5%Mn	97.87	0.51	1.62
Mg0.5%Ca2%Mn	97.54	0.61	1.85
Mg0.5%Ca3%Mn	96.90	0.51	2.59

**Table 3 materials-14-07262-t003:** The average grains dimension of the Mg-0.5Ca-xMn alloys.

Alloys	Average Grains Size [µm]
Mg0.5%Ca0.5%Mn	198 ± 56
Mg0.5%Ca1%Mn	148 ± 15
Mg0.5%Ca1.5%Mn	140 ± 36
Mg0.5%Ca2%Mn	118 ± 31
Mg0.5%Ca3%Mn	79 ± 24

**Table 4 materials-14-07262-t004:** Tissue reactions: Mg-0.5Ca-xMn alloy implants’ clinical exam results.

	The Lumbar Region					The Femoral Region				
Alloys	1 day	7 days	14 days	30 days	60 days	1 day	7 days	14 days	30 days	60 days
M1	M	S	S	A	A	M	S	S	A	A
M2	M	M	S	S	A	M	M	S	S	A
M3	L	M	S	A	A	M	S	S	A	A
M4	M	S	M	S	A	L	M	S	S	A
M5	M	S	M	A	A	L	M	M	S	A

## Data Availability

Not applicable.

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
