# Peer review of "Novel Mg-0.5Ca-xMn Biodegradable Alloys Intended for Orthopedic Application: An In Vitro and In Vivo Study"

_materials, 2021, doi:10.3390/ma14237262_

Round 1
Reviewer 1 Report
Manuscript: Materials 1338052
Title: Novel Mg-0.5Ca-x-Mn biodegradable alloys intended for orthopaedic application: an in vitro and in vivo study
Comments:
The authors studied the microstructure and biocompatibility of Mg-Ca-Mn alloys using in-vitro and in-vivo tests. The topic is interesting and there are aspects of the manuscript which are promising, but the overall quality of the paper needs to be improved before it can be considered for publication. I provide a detailed list of suggestions for the authors below:
- Abstract: The authors refer to Mn alloys, which is incorrect or misleading. There are several spelling mistakes, so please review it carefully.
- Introduction and literature review: The authors refer to several related manuscripts but failed to identify the gaps in the literature. Please clearly indicate the novel aspects of this study and why are they important.
- Materials and methods - Sample preparation: The authors report the ratio of master alloys used for the casting process, but they do not report the final composition of the ingots. Please include the chemical analyses of the resulting alloys in Table 1. Indicate the 3 dimensions of the ingots: length x width x thickness.
- Materials and methods - biocompatibility tests: The dimensional tolerance of the implants is very high: 10-13 mm and 1.5-2.0 mm. This may affect the results. The Mg-Ca master alloy is not included in the methods, but the authors refer to this alloy in the results. Please, mention all the alloys used in the study.
- Results and discussion - Microstructure: The SEM micrographs and EDX maps are of poor quality. The grinding marks are still visible and it is difficult to see the microstructure. The authors use images of different magnification for different alloys, which makes comparisons difficult. The analysis would benefit from X-ray diffraction data to understand the phase composition.
- Results and discussion - Cell viability: The authors interpret the cell viability results on the basis of the degradation rate of different alloys. Although this is relevant, the authors do not have any corrosion data (electrochemical and/or immersion corrosion) to support this statement. As a result, the comment is speculative and does not support the conclusions. Please include corrosion results to support this statement.
- Results and discussion - CT scans and histological analyses: The results are very interesting but they are presented in an inconsistent manner. The authors do not indicate the alloy composition corresponding to each image and the micrographs often have different magnification. Therefore, it is difficult to compare the results and draw conclusions. The SEM images are presented with very little discussion or interpretation. Please expand.
- Conclusions: The authors refer to manganese alloys once again, which is incorrect or misleading. Although the alloys contain manganese, they are essentially magnesium alloys. Please correct this for clarity.
- Conclusions: The observations regarding the degradation rate of the alloys are not sufficiently supported by the experimental results. Please rephrase this or include electrochemical an/or corrosion tests to support this statement.
- English: The manuscript contains several spelling mistakes and typographical errors; it would benefit from a careful review.
Author Response
Dear reviewer,
Thank you very much for your positive feedback. You ask to clarify more details on the magnesium alloys, which certainly would give an improvement to our research. So, thank you for these valuable comments. We have tried to answer all aspects that were mentioned.
Point 1: • Abstract: The authors refer to Mn alloys, which is incorrect or misleading. There are several spelling mistakes, so please review it carefully.
Response 1: The abstract has been revised accordingly
Point 2: • Introduction and literature review: The authors refer to several related manuscripts but failed to identify the gaps in the literature. Please clearly indicate the novel aspects of this study and why are they important.
Response 2:
We have updated the text with the following paragraph:
In the industrial domain, several discoveries have been made involving magnesium-based alloys, but the pure magnesium alloy presents high activity in an aqueous environment, low formability [3,4], low mechanical strength and low precipitation of solid solution, due to the hexagonal crystalline structure [4]. These disadvantages could lead to implant mass integrity loss or H2 excessive release, which can decrease cellular development and can appear in local cysts [5].
- Cao, F; Shi, Z; Song, G-L; Liu, M; Dargusch, M.S.; Atrens, A. Influence of hot rolling on the corrosion behavior of several Mg–X alloys. Corrosion Science 2015, 90, 176-191.
- Bahmani, A; Arthanari, S.; Shin K.S. Improved corrosion resistant and strength of a magnesium alloy using multi-directional forging (MDF). Int. J. Adv. Manuf.Technol. 2019, 105 (1–4), 785–797.
- Haynes, W.M. Properties of the Elements and Inorganic Compounds. CRC Handbook of Physics 2014, 4, 145.
Regarding the novelty, in the text is written the following paragraph:
”In this research paper, the authors studied a novel Mg–0.5 wt.% Ca experimental material which was alloyed with a controlled content of 0.5, 1.0, 1.5, 2.0 and 3.0 wt.% Mn in order to improve the biodegradability rate needed for a symmetrical balance between alloy degradation and bone healing, followed by in vitro 3-(4,5-dimethyltiazol-2-yl)-2,5-diphenyl tetrazolium bromide (MTT) tests to determine its biocompatibility”. The aim of the present work is to study the cellular viability and in vivo implantations, in order to highlight the alloy’s biocompatibility.
Point 3: • Materials and methods - Sample preparation: The authors report the ratio of master alloys used for the casting process, but they do not report the final composition of the ingots. Please include the chemical analyses of the resulting alloys in Table 1. Indicate the 3 dimensions of the ingots: length x width x thickness.
Response 3:
We have inserted in the text the table with chemical compositions determined by EDS analysis. The scan was performed in minimum 5 areas.
The experimental chemical composition is shown in table 2
|
Alloy |
Wt.% Mg |
Wt. %Ca |
Wt. % Mn |
|
Mg-0.5Ca-0.5Mn |
98,89 |
0,56 |
0,55 |
|
Mg-0.5Ca-1Mn |
98,18 |
0,83 |
0,98 |
|
Mg-0.5Ca-1.5Mn |
97,87 |
0,51 |
1,62 |
|
Mg-0.5Ca-2Mn |
97,54 |
0,61 |
1,85 |
|
Mg-0.5Ca-3Mn |
96,90 |
0,51 |
2,59 |
The elaboration process was performed in graphite crucibles and due to the fact that the melting takes place at 675-700 ᵒC, carbon contamination is avoided and also the mini ingots were processed by mechanical processing in order to eliminate surface defects. The melting resulted in mini ingots with a diameter of 20 mm and a height of approximately 50 mm. Samples were cut from these ingots at a thickness of about 2 mm from which the experimental samples were made.
Point 4: • Materials and methods - biocompatibility tests: The dimensional tolerance of the implants is very high: 10-13 mm and 1.5-2.0 mm. This may affect the results. The Mg-Ca master alloy is not included in the methods, but the authors refer to this alloy in the results. Please, mention all the alloys used in the study.
Response 4:
The implants were cut by hand with a orthopaedic plier/bone pncher, then smeared with emery. The dimensions are correlated with the weight of the rats (250-350g), representing a third of the size of the rat femur.
Point 5: •Results and discussion - Microstructure: The SEM micrographs and EDX maps are of poor quality. The grinding marks are still visible and it is difficult to see the microstructure. The authors use images of different magnification for different alloys, which makes comparisons difficult. The analysis would benefit from X-ray diffraction data to understand the phase composition.
Response 5:
SEM images were readjusted and highlighted in Figure 2. X-ray diffraction and light microscopy images were highlighted in another previous research on the same alloy system (1). We mention that this system of alloys is part of studies of a national research contract with a larger research team. XRD analysis shows us three major phases: α-Mg, Mg2Ca and Mg0.975Mn0.025, the previously specified compounds resulting from the peaks of the graph in Figure 3. It is shown that Mg0.975Mn0.025 has the same hexagonal crystallographic structure as pure Mg and Mg2Ca has a monoclinic structure.
Figure 2. Scanning electron images of Mg-0.5Ca-xMn alloys: a) 0.5Mn; b) 1Mn; c) 1.5Mn; d) 2Mn; e) 3Mn.
Figure: XRD analysis of experimental alloys Mg-0.5Ca-xMn (31)
- Oprisan, B.; Vasincu, D.; Lupescu, S.; Munteanu, C.; Istrate, B.; Popescu, D.; Condratovici, C.P.; Dimofte, A.R., Earar K. Electrochemical analysis of some biodegradable Mg-Ca-Mn alloys. Revista de Chimie 2019, 70, 12, 4525 – 4530.
At your suggestion it has been performed the calculation of the average grain sizes of Mg-0.5Ca-xMn alloys. There is a substantial decrease in grain size, with an increase in the concentration of Mn in Mg-0.5Ca-xMn alloys. The values are highlighted in table 3.
Table 3. The average grains dimension of Mg-0.5Ca-xMn alloys.
|
Alloys |
Average grains dimension[µm] |
|
Mg-0.5Ca-0.5Mn |
198 ±56 |
|
Mg-0.5Ca-1Mn |
148 ±15 |
|
Mg-0.5Ca-1.5Mn |
140 ±36 |
|
Mg-0.5Ca-2Mn |
118 ±31 |
|
Mg-0.5Ca-3Mn |
79 ±24 |
Regarding the map’s distribution of elements, a PDF file with these images is offered by the equipment.
Point 6: • Results and discussion - Cell viability: The authors interpret the cell viability results on the basis of the degradation rate of different alloys. Although this is relevant, the authors do not have any corrosion data (electrochemical and/or immersion corrosion) to support this statement. As a result, the comment is speculative and does not support the conclusions. Please include corrosion results to support this statement.
Response 6: The author comments are not speculative at all, and are based on the previously published data[31] concerning the electro-corrosion resistance of the studied alloys, shoving that the corrosion resistance increased with the amount of Mn as alloying element. The article text was modified accordingly.
Point 7: •Results and discussion - CT scans and histological analyses: The results are very interesting but they are presented in an inconsistent manner. The authors do not indicate the alloy composition corresponding to each image and the micrographs often have different magnification. Therefore, it is difficult to compare the results and draw conclusions. The SEM images are presented with very little discussion or interpretation. Please expand.
Response 7: CT images and histological analysis are representative of postoperative times for all studied alloys, macroscopic and microscopic changes due to the presence of implants were followed over time.
7 days after implantation, SEM images showed cell adhesions to the material used. Macrophages, fibroblasts, and giant cells are visible in the fields. An intense neo-synthesis of collagen around the implantation region is also observed (Fig. 9). At 14 days postoperatively, we further identified cell adhesions and neoformation collagen fibers. At the same time, we noticed the presence of inflammatory reactions, lymphocytes, and macrophages on the surface of the material used. The adhesion to the material of giant cells and macrophages, involved in the process of resorption and proliferation of fibroblasts and the formation of collagen (Fig. 10) is also observed. After a month (30 days), it was also observed adhesions with giant cells and macrophage, involved in the resorption of the material, proliferation with fibroblasts, and the synthesis of collagen fibers. From the images, we can conclude that the connective tissue of neogenesis replaced most of the resorbed material (Fig. 11). At 60 days, the SEM images show a new area of connective tissue that replaced the material, so total resorption (disappearance) of the alloy (implant). (Fig. 12.)
The results of histological and electro-microscopic examinations are identical, confirming an efficient bio-integration of the studied material. We can observe on examination after 60 days, that the number of collagen fibers is much higher, compared to the period of 7, 14, and 30 days, indicating good biocompatibility, which ensures the recommendation to be used in osteosynthesis/orthopedics. A large number of collagen fibers around the implant and the high cellularity (fibroblasts, giant Muller cells, macrophages, histiocytes, lymphocytes) indicate an intense resorption process and good tissue compatibility. No areas of peri-implant necrosis were noticed, which, in our opinion, indicates an excellent tolerance of the material, making it ideal for use as an osteosynthesis material.
Point 8: • Conclusions: The authors refer to manganese alloys once again, which is incorrect or misleading. Although the alloys contain manganese, they are essentially magnesium alloys. Please correct this for clarity.
Response 8: The paragraph has been corrected
Point 9: • Conclusions: The observations regarding the degradation rate of the alloys are not sufficiently supported by the experimental results. Please rephrase this or include electrochemical an/or corrosion tests to support this statement.
Response 9: Electrochemical studies performed in another experimental research [31] on the same set of samples, presented that the highest corrosion rate is on the alloy with 0.5% Mn respectively 0.85 mm / year and the smallest corrosion rate is for 3% Mn, respectively 0.55 mm / year. It follows that increasing the concentration of the Mn alloying element in the Mg-0.5Ca-xMn system leads to a decrease in the biodegradation rate. The electrochemical polarization curves indicate for all samples a generalized surface corrosion more pronounced in the case of samples with 1 and 2% Mn respectively. In all cases there is seen a combined corrosion between galvanic, intercrystalline and point corrosion.
Point 10: • English: The manuscript contains several spelling mistakes and typographical errors; it would benefit from a careful review.”
Response 10: The authors tried to revise the spelling mistakes and the typographical errors. We hope it is ok now.
- Oprisan, B.; Vasincu, D.; Lupescu, S.; Munteanu, C.; Istrate, B.; Popescu, D.; Condratovici, C.P.; Dimofte, A.R., Earar K. Electrochemical analysis of some biodegradable Mg-Ca-Mn alloys. Revista de Chimie 2019, 70, 12, 4525 – 4530.

Reviewer 2 Report
Notes on the article of Corneliu Munteanu, Daniela Maria Vlad, Eusebiu-Viorel Sindilar, Bogdan Istrate, Maria Butnaru, Sorin Aurelian Pasca, Roxana Oana Nastasa, Iuliana Mihai and Stefan-Lucian Burlea “Novel Mg-0.5Ca-xMn biodegradable alloys intended for orthopedic application: An in vitro and in vivo study”
The paper reports results of studying of the structure and biocompatibility (both in vitro and in vivo) of the alloys Mg-0.5%Ca-xMn. The authors have provided extensive and carefully done biocompatibility in vitro and in vivo researches, but the materials science part (including the description of aspects of the microstructure and the study of the corrosion properties of alloys) seems to be quite poor. However, the results of this article have the high importance for future studies and design of medical devices based on magnesium alloys. This is an interesting and well-written report, which should be published after major revisions that are listed below:
1) It is necessary to give a more detailed description of the microstructure (the sizes of the structural elements and the volume of the phases).
2) The authors should provide a section where they give the results of the studying of the degradation behaviour of the alloys. Biodegradation rate values are needed to understand the effect of the alloy composition on biocompatibility and volume release of hydrogen.
3) Figure 8 shows the wrong labeling of the images.
4) Следующие опечатки должны быть исправлены:
- P.1, L. 40: “Mg alloy” instead of “Mn alloy”.
- P.2, L. 65: “…the lamellar eutectic compound Mg2Ca…” instead of “…the lamellar eutectic compound Mg2Ca…”.
- P.4, L. 169: “…for 30 min at 37 °C…” instead of “…for 30 min at 37oC…”.
- P.4, L. 180: “…having 10-13 mm length and 1.5-2 mm height…” instead of “…having 10-13 mm length and 1,5-2mm height…”.
- P.8, L. 277: “…by reduction of H2O to H2 gas and OH−, ions and Mg(OH)2 as corrosion product…” instead of “…by reduction of H2O to H2 gas and OH−, ions and Mg(OH)2 as corrosion product…”.
- P. 12, L. 377: “…a very good resorption / biodegradation of the implanted material and a moderate inflammatory reaction” instead of “…a very good resorption / biodegradation of the implanted material and a and a moderate inflammatory reaction”.
- P. 12, L. 380 and 386, P. 13, L. 394 and 401: “…Mg-0.5Ca-xMn alloys …” instead of “…Mg-0,5Ca-xMn alloys …”.
- P. 14, L. 422: “Degradation of Mg alloy implants…” instead of “Degradation of Mn alloy implants…“ (or the authors should use “Mn-containing alloy”).
Author Response
Dear reviewer,
Thank you very much for your positive feedback. You ask to clarify more details on the magnesium alloys, which certainly would give an improvement to our research. So, thank you for these valuable comments. We have tried to answer all aspects that were mentioned.
Point 1: It is necessary to give a more detailed description of the microstructure (the sizes of the structural elements and the volume of the phases).
Response 1: SEM images were readjusted and highlighted in Figure 2. X-ray diffraction and light microscopy images were highlighted in another previous research on the same alloy system [31]. XRD analysis and light microscopy performed by Oprisan et al. [31] on the same set of samples presented three major phases: α-Mg, Mg2Ca and Mg0.975Mn0.025. It is shown that Mg0.975Mn0.025 has the same hexagonal crystallographic structure with pure Mg and Mg2Ca has a monoclinic structure.
Figure 2. Scanning electron images of Mg-0.5Ca-xMn alloys: a) 0.5Mn; b) 1Mn; c) 1.5Mn; d) 2Mn; e) 3Mn.
Figure: XRD analysis of experimental alloys Mg-0.5Ca-xMn (31)
The calculation of the average grain sizes of Mg-0.5Ca-xMn alloys has been measured in minimum of 7 areas. It has been observed that a substantial decrease in grain size appeared, due to the increase in the concentration of Mn in Mg-0.5Ca-xMn alloys. The values are highlighted in table 3.
Table 3. Average grains dimension
|
Alloys |
Average grains dimension[µm] |
|
Mg-0.5Ca-0.5Mn |
198 ±56 |
|
Mg-0.5Ca-1Mn |
148 ±15 |
|
Mg-0.5Ca-1.5Mn |
140 ±36 |
|
Mg-0.5Ca-2Mn |
118 ±31 |
|
Mg-0.5Ca-3Mn |
79 ±24 |
We also have inserted in the text the table with chemical compositions determined by EDS analysis. The scan was performed in minimum 5 areas.
The experimental chemical composition is shown in table 2.
|
Alloy |
Wt.% Mg |
Wt. %Ca |
Wt. % Mn |
|
Mg-0.5Ca-0.5Mn |
98,89 |
0,56 |
0,55 |
|
Mg-0.5Ca-1Mn |
98,18 |
0,83 |
0,98 |
|
Mg-0.5Ca-1.5Mn |
97,87 |
0,51 |
1,62 |
|
Mg-0.5Ca-2Mn |
97,54 |
0,61 |
1,85 |
|
Mg-0.5Ca-3Mn |
96,90 |
0,51 |
2,59 |
Point 2: he authors should provide a section where they give the results of the studying of the degradation behaviour of the alloys. Biodegradation rate values are needed to understand the effect of the alloy composition on biocompatibility and volume release of hydrogen.
Response 2: The Mg-0.5Ca-xMn alloy system is part of the larger research of a national project. Electrochemical studies performed in another experimental research [31] on the same set of samples, presented that the highest corrosion rate is on the alloy with 0.5% Mn respectively 0.85 mm / year and the smallest corrosion rate is for 3% Mn, respectively 0.55 mm / year. It follows that increasing the concentration of the Mn alloying element in the Mg-0.5Ca-xMn system leads to a decrease in the biodegradation rate. The electrochemical polarization curves indicate for all samples a generalized surface corrosion more pronounced in the case of samples with 1 and 2% Mn respectively. In all cases there is seen a combined corrosion between galvanic, intercrystalline and point corrosion.
Point 3: Figure 8 shows the wrong labeling of the images.
Response 3: We corrected the labeling of the images and pointed with arrows in the description.
Point 4: Следующие опечатки должны быть исправлены:
- P.1, L. 40: “Mg alloy” instead of “Mn alloy”. - corrected
- P.2, L. 65: “…the lamellar eutectic compound Mg2Ca…” instead of “…the lamellar eutectic compound Mg2Ca…”.- corrected
- P.4, L. 169: “…for 30 min at 37 °C…” instead of “…for 30 min at 37oC…”.- corrected
- P.4, L. 180: “…having 10-13 mm length and 1.5-2 mm height…” instead of “…having 10-13 mm length and 1,5-2mm height…”.- corrected
- P.8, L. 277: “…by reduction of H2O to H2 gas and OH−, ions and Mg(OH)2 as corrosion product…” instead of “…by reduction of H2O to H2 gas and OH−, ions and Mg(OH)2 as corrosion product…”.- corrected
- P. 12, L. 377: “…a very good resorption / biodegradation of the implanted material and a moderate inflammatory reaction” instead of “…a very good resorption / biodegradation of the implanted material and a and a moderate inflammatory reaction”. - corrected
- P. 12, L. 380 and 386, P. 13, L. 394 and 401: “…Mg-0.5Ca-xMn alloys …” instead of “…Mg-0,5Ca-xMn alloys …”.- corrected
- P. 14, L. 422: “Degradation of Mg alloy implants…” instead of “Degradation of Mn alloy implants…“ (or the authors should use “Mn-containing alloy”).” - corrected
Response 4: All the aspects have been corrected.

Reviewer 3 Report
Review of the manuscript "Novel Mg-0.5Ca-xMn biodegradable alloys intended for orthopedic application: An in vitro and in vivo study".
The manuscript is interesting, almost clear, balanced and well organized. However the main lack is english language. The manuscript must deeply revised by a native-speaker (just as example, line 20: "materials... has been...").
After that it can be reconsidered for publication.
Author Response
Dear reviewer,
Thank you very much for your positive feedback. You ask to clarify more details on the magnesium alloys, which certainly would give an improvement to our research. We have tried to correct the English as you mentioned and we hope now is better.

Round 2
Reviewer 1 Report
I thank the authors for the responses and for the efforts to improve the manuscript. The manuscript has indeed improved in its presentation, but my main comments remain valid (see below). Therefore, I consider the article is not acceptable for publication in its present format and I suggest the authors to review it carefully before resubmitting it.
1- Literature review: The authors refer to several relevant publications on Mg alloys containing Ca and Mn, but the gaps in the knowledge are not clearly stated. Please clearly indicate the novel aspects of this work (is this related with the ternary Mg-Ca-Mn alloys? Is there another publication by the same authors on the microstructure and properties of these alloys? Why are these alloys particularly interesting over other alloy compositions? Do the biocompatibility tests provide new information that other manuscripts do not cover?).
2- Materials and methods - biocompatibility tests: The dimensional tolerance of the implants is very high (10-13 mm and 1.5-2.0 mm) and may affect the results. In their response, the authors indicated that the dimensions of the implant were correlated with the weight of the rats (250-350g), and the size of the rat femur, but this information was not included in the manuscript. Please indicate how the size of the implant was correlated with the weight of the rats and the size of the femur.
3- Results and discussion - Microstructure: The SEM micrographs and EDX maps are of poor quality. The grain sizes reported in Table 3 do not correlate with the micrographs in Figure 2. The EDX maps do not show clear differences and the white precipitates do not reflect a higher Mn content as described in the text. Please review these results and improve them accordingly. The manuscript would benefit from XRD data. If possible, the authors should present the relevant information in this manuscript, rather than citing results from other manuscripts (Ref 31).
4- Results and discussion - Cell viability: The authors interpret the cell viability results on the basis of the chemical stability and degradation rate of different alloys. However, the degradation rate was not measured or assessed in this study. The authors discuss their findings in the context of relevant literature, but the evidence is not conclusive. Therefore, the comments on the chemical stability of Mg-Ca-Mn alloys are not well supported by the results presented in this manuscript.
5- Results and discussion - CT scans and histological analyses: The results are very interesting but they are presented in an inconsistent way. The authors do not indicate the alloy composition corresponding to each image and the micrographs often have very different magnification, which makes comparisons difficult. Table 4 (incorrect number) on page 10 indicates that the alloys responded differently after implantation (H2 release), but this is not reflected in the subsequent results. Please expand and discuss this further.
6- Conclusions: The observations regarding the degradation rate of the alloys are not sufficiently supported by the experimental results. Please rephrase this or include electrochemical an/or corrosion tests to support this statement. The authors should present the relevant information in this manuscript, rather than citing results from other manuscripts.
Author Response
Dear reviewer,
Thank you very much for your positive feedback. You ask to clarify more details on the magnesium alloys, which certainly would give an improvement for our research. So, thank you for these valuable comments. We have tried to answer to all aspects that were mentioned.
Point 1: Literature review: The authors refer to several relevant publications on Mg alloys containing Ca and Mn, but the gaps in the knowledge are not clearly stated. Please clearly indicate the novel aspects of this work (is this related with the ternary Mg-Ca-Mn alloys? Is there another publication by the same authors on the microstructure and properties of these alloys? Why are these alloys particularly interesting over other alloy compositions? Do the biocompatibility tests provide new information that other manuscripts do not cover?).
Response 1: The introduction highlights articles with high impact that address the issue of biodegradable materials based on Mg-Ca-Mn. The main author (Prof. Munteanu) and Prof.Bogdan Istrate participated as co-authors in a paper in which are presented only some aspects of microstructure (optical microstructure and XRD analysis-ref 31) and also aspects related to corrosion results.
The results of this paper are original and come from the collaboration carried out within a complex national project with 10 partners [7 universities (3 with medical faculties and one with veterinary medicine faculty) and 3 national RD research institutes] in which they were analysed 4 material classes (Mg-Ca-Zr / Y / Mn / Gd). The Mg-Ca-Mn system performed best in terms of cell viability and as well as in vivo experiments, which are presented in the present paper. The published results for the other mentioned systems have been published and can be analyzed in the following references 25, 29 and *:
- Istrate, B.; Munteanu, C.; Cimpoesu, R.; Cimpoesu, N.; Popescu, O.D.; Vlad, M.D. Microstructural, Electrochemical and In Vitro Analysis of Mg-0.5Ca-xGd Biodegradable Alloys. Appl. Sci. 2021, 11, 981, Q2
- Sindilar, E.-V.; Munteanu, C.; Pasca, S.A.; Mihai, I.; Henea, M.E.; Istrate, B. Long Term Evaluation of Biodegradation and Biocompatibility In-Vivo the Mg-0.5Ca-xZr Alloys in Rats. Crystals 2021, 11, 54, Q2
* Istrate, B.; Munteanu, C.; Lupescu, S.; Chelariu, R.; Vlad, M.D.; Vizureanu, P. Electrochemical Analysis and In Vitro Assay of Mg-0.5Ca-xY Biodegradable Alloys. Materials 2020, 13, 3082. https://doi.org/10.3390/ma13143082, Q1
Point 2: Materials and methods - biocompatibility tests: The dimensional tolerance of the implants is very high (10-13 mm and 1.5-2.0 mm) and may affect the results. In their response, the authors indicated that the dimensions of the implant were correlated with the weight of the rats (250-350g), and the size of the rat femur, but this information was not included in the manuscript. Please indicate how the size of the implant was correlated with the weight of the rats and the size of the femur.
Response 2: We calculated the dimensions of the implants using a specific orthopaedic algorithm. The weight of the animal is directly proportional to the size of the long bones.
- The length of the implant was set at one third of the length of the femur.
- The thickness of the implant was calculated taking into account the average diameter in the middle third and the fact that it must not exceed 60% of the bone diameter.
The table presents some measurements that were the basis for establishing the size of the implants:
|
Rat weight (g) |
Femur diameter (mm) |
Implant diameter (mm) |
Femur diameter * 0.6 = Implant diameter (mm) |
Femur Length (mm) |
Implant Length (mm) |
Femur Length/ 0.33= Implant Length (mm) |
|
350 |
3.30 |
2.0 |
1.98 |
37.9 |
13 |
37.9/0.33=12.633 |
|
338 |
3.10 |
1.9 |
1.86 |
36.6 |
12 |
36.6/0.33=12.200 |
|
330 |
2.80 |
1.7 |
1.68 |
36.5 |
12 |
36.5/0.33=12.167 |
|
292 |
2.71 |
1.7 |
1.62 |
32.3 |
11 |
32.3/0.33=10.767 |
|
284 |
2.67 |
1.6 |
1.60 |
31.4 |
11 |
31.4/0.33=10.467 |
|
271 |
2.58 |
1.5 |
1.55 |
30.9 |
10 |
30.9/0.33=10.300 |
|
262 |
2.23 |
1. 5 |
1.5 |
29.7 |
10 |
29.7/0.33=9.900 |
Point 3: Results and discussion - Microstructure: The SEM micrographs and EDX maps are of poor quality. The grain sizes reported in Table 3 do not correlate with the micrographs in Figure 2. The EDX maps do not show clear differences and the white precipitates do not reflect a higher Mn content as described in the text. Please review these results and improve them accordingly. The manuscript would benefit from XRD data. If possible, the authors should present the relevant information in this manuscript, rather than citing results from other manuscripts (Ref 31).
Response 3: The authors improved the quality of the SEM figures and added the figures with the measurements for correlation with Table 3. The quality of the distribution maps has been improved, and at present the chemical distribution of all elements can be distinguished. The X-ray diffraction figure was also introduced where the phases and compounds formed are highlighted.
Figure 2.
Figure 3
Figure 4
Point 4: Results and discussion - Cell viability: The authors interpret the cell viability results on the basis of the chemical stability and degradation rate of different alloys. However, the degradation rate was not measured or assessed in this study. The authors discuss their findings in the context of relevant literature, but the evidence is not conclusive. Therefore, the comments on the chemical stability of Mg-Ca-Mn alloys are not well supported by the results presented in this manuscript.
Response 4: The author comments are based on the previously published data concerning the electro-corrosion resistance of the same studied alloys, shoving that the corrosion resistance increased with the amount of Mn as alloying element. Indeed, there is another publication (on the microstructure and properties of these Mg-Ca-Mn alloys) by some of the authors of the present work.
[31] Oprisan, B.; Vasincu, D.; Lupescu, S.; Munteanu, C.; Istrate, B.; Popescu, D.; Condratovici, C.P.; Dimofte, A.R., Earar K. Electrochemical analysis of some biodegradable Mg-Ca-Mn alloys. Revista de Chimie 2019, 70, 12, 4525 – 4530.
In our experimental set-up of the present work was not included the in vitro quantification of the alloys’ degradation rate, due to the fact that all of the physico-chemical processes taking place during the degradation process of Mg-alloys depend on the microenvironment condition and, from this point of view, a different in vivo corrosion/biodegradation scenario (i.e. both composition and flow of biological fluids) of Mg-alloys take place under in vivo applications that may influence the in vivo alloys’ biodegradation rate and subsequent resorption (as was demonstrated by the in vivo study). Therefore, the comments on the chemical stability/degradability of Mg-Ca-Mn alloys are supported by our previously published results concerning the Mg-Ca-Mn alloys’ corrosion and by the in vivo study results.
The article text was modified accordingly:
In addition, these facts are supported by the our previously published data [31] concerning the electro-corrosion resistance of the same studied alloys, shoving that the corrosion resistance increased with the amount of Mn as alloying element. Briefly, the electrochemical studies performed on the same set of samples [31], evidenced that the highest corrosion rate resulted for the alloy with 0.5% Mn respectively 0.85 mm / year, and the smallest corrosion rate resulted for 3% Mn, respectively 0.55 mm / year, leading to a decrease in the biodegradation rate.
Point 5: Results and discussion - CT scans and histological analyses: The results are very interesting but they are presented in an inconsistent way. The authors do not indicate the alloy composition corresponding to each image and the micrographs often have very different magnification, which makes comparisons difficult. Table 4 (incorrect number) on page 10 indicates that the alloys responded differently after implantation (H2 release), but this is not reflected in the subsequent results. Please expand and discuss this further.
Response 5: We corrected Table no 4.
Table 4 shows the observations of the clinical examination, the changes visible to the naked eye due to gas bags, were evaluated as: large (L), medium (M), small (S) and absent (A).
Through tomography and histological analysis, we wanted to highlight the presence of gas bags at 60 days, which could not be found at the clinical examination.
We completed with:
At 7 days, medium gas accumulations with deformations of the region were observed in groups M2, M3, M4 and M5 (Figure 7.a, 7.b and Figure 8.a, 8.b) and small gas accumulations in the group M1. At 14 days, medium gas accumulations were observed in the femoral region in groups M4 and M5 (Figure 7.c, 7.d and Figure 8.c, 8.d) and small accumulations of gas in groups M1, M2 and M3. At 30 days, small accumulations of gas with deformations of the region in groups M2, M4 and M5 and small accumulations of gas without deformation in groups M1 and M3 were observed (Figure 7.e, 7.f and Figure 8.e, 8.f). At 60 days, small gas accumulations without deformations of the region were observed in all groups (Figure 7.g, 7.h and Figure 8.g, 8.h).
Imaging observations revealed the lowest amount of H2 for M1 and M3.
Figure 9. Representative results of histological exams of MTC stained sections of rat tissues after Mg-0.5Ca-xMn alloy implantation
Point 6: 6- Conclusions: The observations regarding the degradation rate of the alloys are not sufficiently supported by the experimental results. Please rephrase this or include electrochemical an/or corrosion tests to support this statement. The authors should present the relevant information in this manuscript, rather than citing results from other manuscripts.”
Response 6: The text regarding the electrochemical analysis was removed from the section "Conclusions" and was integrated in the chapter "3.2. Cell viability ”.

Reviewer 2 Report
The authors carefully revised the article and answered all questions. I recommend to accept the revised version of the article in the present form.
Author Response
Dear reviewer,
Thank you very much for your positive feedback and acceptance of the article for publishing.
Reviewer 3 Report
The manuscript has been significatively improved and can be accepted in the present form.
Author Response

(The authors gave the same response as above.)
